# Architecture of the *Saccharomyces cerevisiae* NuA4/TIP60 complex

Xuejuan Wang [1,2], Salar Ahmad[3], Zhihui Zhang[1], Jacques Côté [3] & Gang Cai [1,2]

The NuA4/TIP60 acetyltransferase complex is required for gene regulation, DNA repair and cell cycle progression. The limited structural information impeded understanding of NuA4/TIP60 assembly and regulatory mechanism. Here, we report the 4.7 Å cryo-electron microscopy (cryo-EM) structure of a NuA4/TIP60 TEEAA assembly (Tra1, Eaf1, Eaf5, actin and Arp4) and the 7.6 Å cryo-EM structure of a TEEAA-piccolo assembly (Esa1, Epl1, Yng2 and Eaf6). The Tra1 and Eaf1 constitute the assembly scaffold. The Eaf1 SANT domain tightly binds to the LBE and FATC domains of Tra1 by ionic interactions. The actin/Arp4 peripherally associates with Eaf1 HSA domain. The Eaf5/7/3 (TINTIN) and piccolo modules largely pack against the FAT and HEAT repeats of Tra1 and their association depends on Eaf1 N-terminal and HSA regions, respectively. These structures elucidate the detailed architecture and molecular interactions between NuA4 subunits and offer exciting insights into the scaffolding and regulatory mechanisms of Tra1 pseudokinase.

[1] Hefei National Laboratory for Physical Sciences at Microscale and School of Life Sciences, University of Science & Technology of China, Hefei 230026, China. [2] CAS Center for Excellence in Molecular Cell Science, Chinese Academy of Sciences, Hefei 230026, China. [3] St-Patrick Research Group in Basic Oncology, Laval University Cancer Research Center, CHU de Québec Research Center-Oncology Axis, Quebec City QC G1R 3S3, Canada. Correspondence and requests for materials should be addressed to X.W. (email: xuejuan@ustc.edu.cn) or to G.C. (email: gcai@ustc.edu.cn)

Acetylation is one such post-translational modification that critically regulates chromatin function[1]. Histone acetyltransferases (HATs) often exist in large multimeric complexes, such as the deeply conserved NuA4 (Nucleosome acetyltransferase of H4)[2,3] and SAGA (Spt-Ada-Gcn5-Acetyltransferase)[4,5]. The NuA4 complex plays essential roles in crucial genomic processes including DNA damage repair and transcription[6–10]. The catalytic subunit Esa1/Tip60/KAT5 along with five non-catalytic NuA4 subunits: Epl1 (EPC1/2 ortholog), Tra1 (TRRAP ortholog), Arp4 (BAF53), Act1, and Swc4/Eaf2 (DMAP1), are essential in *Saccharomyces cerevisiae* and broadly conserved[11]. Besides well-established histones substrates, NuA4 complex acetylates more than 250 non-histone substrates, critically controlling metabolism, autophagy, and homeostasis[12–14]. In humans, the essential catalytic subunit Tip60, along with several other NuA4 subunits including TRRAP, are associated with tumorigenesis in different cancers such as colon, breast, and prostate tumors and are essential for stem cell maintenance and renewal[15,16].

NuA4 is a large complex comprising of 13 unique subunits in yeast with a combined molecular weight of 1.0 MDa. A hallmark of the NuA4 complex is that it shares subunits with other protein complexes involved in chromatin-binding and histone modification reactions. There are at least two independent NuA4 sub-complexes that exist in vivo: piccolo-NuA4, composed of Esa1, Epl1, Yng2, and Eaf6[17,18], and the TINTIN triad of Eaf5/7/3[9,19]. Many other NuA4 subunits participate in several distinct chromatin modifying/remodeling complexes. These include Tra1 (TRRAP), which serves as a recruitment module in SAGA and SLIK/SALSA complexes[20], and Swc4/Eaf2, Yaf9, Arp4, and Act1, which are components of the SWR1 chromatin remodeling/histone exchange complex[21,22]. In addition, Arp4 and Act1 are also found in the INO80 ATP dependent chromatin-remodeling complex[23] and Eaf3 is also part of the Rpd3S histone deacetylase complex[24,25]. Eaf1 is the sole subunit that is exclusively associated with the NuA4 complex in vivo[6,26]. It serves as a platform that coordinates the assembly of four different functional modules: piccolo-NuA4, Tra1, Eaf3/5/7, and the Arp4/Act1/Swc4/Yaf9.

Tra1 (human TRRAP ortholog) is a highly conserved 3744-residue pseudokinase that belongs to the evolutionarily conserved phosphatidylinositol-3-kinase-related protein kinase (PIKK) family. Each of the six members in the PIKK family plays pivotal roles in controlling cellular homeostasis, including genomic stability (ATM, ATR, and DNA-PKcs), cell growth and metabolism (mTOR), mRNA decay (SMG1), and transcriptional regulation (TRRAP)[27]. A highly conserved C-terminal FAT/kinase/FATC domain architecture[28,29] and an extended N-terminal HEAT repeat domain are shared in the PIKKs[30–32]. Cryo-electron microscopy (cryo-EM) has recently yielded structures of PIKKs that were otherwise unobtainable with traditional approaches. Recently, besides a 4.3 Å crystal structure of DNA-PKcs[33], high resolution cryo-EM structures of ATM/Tel1[34,35], ATR/Mec1[36], and TRRAP/Tra1[37] have recently been reported. Interestingly, Tra1 and DNA-PKcs contain a much larger N-terminal HEAT repeat region than other PIKK members, which are strikingly similar in three-dimensional (3D) architecture, especially in the topology of the HEAT repeats[33,37].

PIKKs generally work as a part of larger assemblies with several accessory and regulatory subunits. These complexes are dynamic in composition and protein–protein and protein–DNA interactions critically regulate the recruitments and functions of PIKKs. The extensive interaction surfaces enable PIKKs to integrate the information provided by multiple accessory subunits and nucleic acids. Interestingly, the non-catalytic properties of mTOR, ATR/Rad3, and ATM/Tel1 are proved to be essential[38]. The large size, lack of kinase

activity, and presence in both NuA4 and SAGA complexes have suggested that Tra1/TRRAP may serve as a scaffold for complex assembly or for recruitment to chromatin[39,40], which offers a unique paradigm illuminating the non-catalytic functions of PIKKs.

However, the understanding of NuA4 and SAGA assembly has been impeded so far by limited structural information. Due to the high composition complexity and conformation flexibility of the NuA4 and SAGA complexes, past structural analyses have been limited to low resolution structures[41–44]. Here, we report the cryo-EM structures of the NuA4 complex from *S. cerevisiae* without the piccolo module (4.7 Å) and including it (7.6 Å). The structures provide detailed information on the interactions between NuA4 subunits and the scaffolding role of Eaf1 and the Tra1 pseudokinase.

## Results

**Purification and negative stain electron microscopy reconstruction.** The previous electron microscopy (EM) studies suggested that both composition and conformation of the SAGA complexes show substantial heterogeneity[41,43]. Surprisingly, the sole cryo-EM reconstruction of the yeast NuA4 complex[44] was recently described as corresponding uniquely to the structure of the Tra1 subunit[37,45]. Towards determining the high-resolution structure of NuA4 complex, we have established an efficient purification procedure to acquire endogenous NuA4 complex directly from yeast cells. The purification involves ammonium sulfate precipitation to enrich the NuA4 containing fraction and one-step affinity chromatography purification using anti-FLAG column. To further remove minor contaminants, an ion exchange Mono Q column was employed. The procedure yielded highly homogeneous NuA4 complex that is uniform in composition with all the 13 subunits (Fig. 1a, adapted from refs. [6,11]) using SDS–PAGE analysis (Supplementary Fig. 1a). The particles observed with EM appeared well preserved under negative stain and were similar to each other in size and overall shape (Supplementary Fig. 1b). Thus, we obtained high-quality NuA4 complex suitable for cryo-EM analysis and avoided chemically crosslinking and heavy metals staining deformations.

First, we performed negative-stain EM analysis to detect the potential heterogeneity of the endogenous NuA4 preparation and obtain an initial model to refine the cryo-EM dataset. Two compositional states were readily identified from the reference-free alignment and classification of negative-stain EM images (Supplementary Fig. 1c). The smaller state I is more predominant than larger state II (1223 vs. 883 particles). To ensure that the different classes indeed represent different composition states of the NuA4 complex and not just different views, we obtained 3D structures of each state from images of tilted stained particles. Both maps reveal a diamond ring shaped density strikingly reminiscent of the Tra1 and DNA-PKcs structures previously described[33,37] (Supplementary Fig. 1c). While the two stained reconstructions may suffer from distortions due to the missing cone and the flattening artefacts, they could still be sufficiently accurate to be used as initial models for refinements. To minimize the effects of the possible distortions, one RCT model was low-pass filtered to 60 Å and used just as the starting model for the 3D classification.

**Cryo-EM structure of the NuA4 TEEAA sub-complex.** High-resolution images were recorded on a Titan Krios TEM equipped with a K2 camera (Supplementary Fig. 2a). Classification of raw cryo-EM particles resulted in well-resolved 2D class averages, with secondary structural features clearly discernable (Fig. 2a and Supplementary Fig. 2b). After 3D classification of particles (Supplementary Fig. 3), a subset of particles corresponding to the

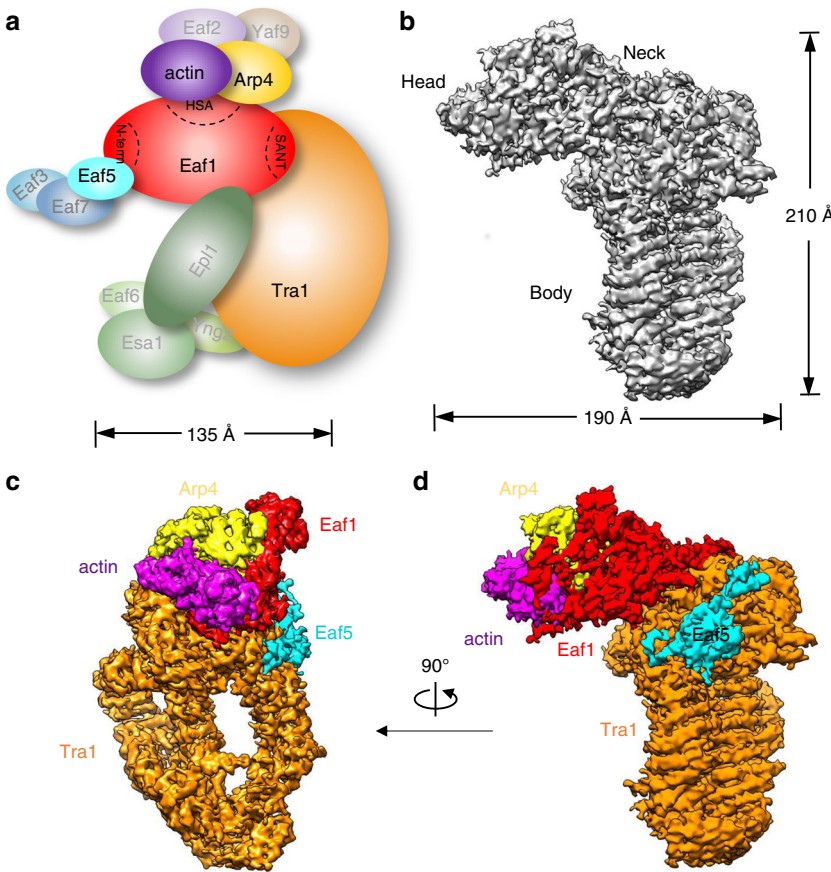

**Fig. 1** Cryo-EM structure of the NuA4 TEEAA sub-complex. **a** Schematic view of subunit and modular organization of the NuA4 complex from *S. cerevisiae*. The TEEAA subunits are highlighted. **b** The front view of the cryo-EM density map of the NuA4 TEEAA assembly at 4.7 Å. **c**, **d** Two views of the NuA4 TEEAA assembly with the density map color-coded by subunit assignment, the same as panel **a**. Arp4 is in yellow, actin in purple, Eaf5 in cyan, Tra1 in orange and Eaf1 in red

state I NuA4 was subjected to high-resolution refinement, resulting in a 3D density map with final overall resolution of 4.7 Å (Gold-standard FSC 0.143 criterion), with higher resolutions in the center and lower resolutions in the periphery (Supplementary Fig. 4 and Supplementary Table 1). The EM density map can be clearly divided into three parts, with an elongated "Head" bridged by the "Neck" to the diamond-ring shaped "Body" (Fig. 1b). The diameter determined from the front view and the axial height of the structure are ~210 and 190 Å, respectively.

We homology modeled the C-terminal FAT-KD-FATC domain of the Tra1 based on the DNA-PKcs crystal structure (PDBID: 5LUQ)[33] and de novo modeled the main chain of the N-terminal HEAT repeats by taking into account of the typical topology of the HEAT repeats. The resulting Tra1 model was real-space refined and further optimized. Our Tra1 structure is well consistent with the 3.7 Å cryo-EM model of the yeast Tra1 (PDB ID: 5OJS) (Supplementary Fig. 5)[37]. Subsequent unambiguously docking of the Tra1 model and the crystal structures of actin-Arp4-Swr1 HSA (PDBID: 5I9E)[46] into the map resulted in high correlation coefficients. The Eaf1 subunit is the assembly platform of the NuA4 complex, which critically depends on its HSA and SANT domains regions[6]. We homology modeled the HSA and SANT domains of Eaf1 and the additional EM density at the Neck region could only accommodate ~400 additional amino acids of the Eaf1, which is largely disordered and could not allow model building at the current resolution (Fig. 1c, d). The dissociable Eaf3/5/7 triad is tethered to the NuA4 complex by Eaf5 subunit[9] and the main chains of Eaf5 could be clearly traced guided by the secondary structure prediction. Therefore, the state

I NuA4 sub-complex contains Tra1, Eaf1, Eaf5, actin, and Arp4. We named the sub-complex as the TEEAA assembly, for the **T**ra1, **E**af1, **E**af5, **a**ctin, and **A**rp4 subunit composition. The NuA4 TEEAA, lacking the piccolo, Eaf3/7 and Swc4/Yaf9, represents the predominant compositional state of the NuA4 complex in our preparations (Fig. 2b). This observation is consistent with the dissociable nature of the piccolo[17,18] and the Eaf3/5/7 modules[9,19] and the sharing of the Swc4/Yaf9 dimer between NuA4 and SWR1 complexes, which has been proposed to recruit both complexes to the chromatin[47,48].

**Structure of Tra1/TRRAP.** The molecular weights of the DNA-PKcs and TRRAP/Tra1 exceed 430 kDa, which contain much larger N-terminal HEAT repeat region than other PIKK members[49]. The HEAT repeats critically regulate PIKK function, which are essential for binding proteins and DNA that associate with the PIKKs to regulate their activity and cellular localization[30]. Similar to the DNA-PKcs[33], Tra1 folds into three well defined structural units: N-terminal unit (residues 1 to 824), the Circular Cradle (residues 825 to 2630), and the C-terminal FAT-KD-FATC domain (residues 2631 to 3744, FAT, FRB, kinase, PRD, and FATC) (Fig. 3a, b). Compared with the DNA-PKcs structure, the HEAT repeats of the Tra1 show the largest variation with the two distinct alpha solenoids drawing near to each other. The N-terminal unit of Tra1 moves inwards by 60 Å, while the circular cradle unit moves a smaller distance by 25 Å (Fig. 3a and Supplementary Fig. 6). The gigantic HEAT repeats in DNA-PKcs and Tra1/TRRAP could accommodate specific functions,

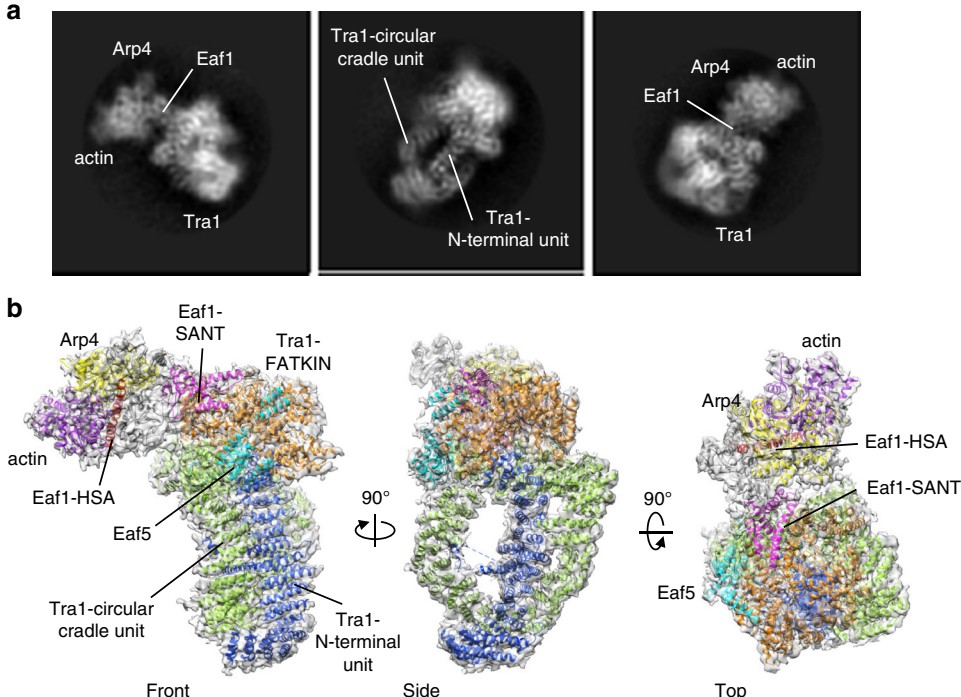

**Fig. 2** Architecture of the NuA4 TEEAA assembly. **a** Three representative 2D class averages of the NuA4 TEEAA. **b** Three views of the NuA4 TEEAA assembly. The cryo-EM density map is shown as a translucent surface and fitted with the ribbon diagram model. Arp4 is in yellow, actin in purple, and Eaf5 in cyan. The Tra1 FATKIN is shown in orange and the N-terminal HEAT repeats are colored in blue (N-terminal unit) and in light green (circular cradle unit). The Eaf1 is colored in red (HSA domain), in magenta (SANT domain), and in gray (other regions). Each successive view is rotated as indicated

such as interactions with DNA and DNA repair factors[33]. These observed conformational differences likely allow the functional divergence of DNA-PKcs and Tra1/TRRAP to bind different DNA fragment or repair factors.

Tra1 contains a canonical two-lobe kinase domain (KD) spanning about 500 C-terminal residues, with four characteristic insertions: FRB (~100 residues), FATC (~30 residues), LBE (~40 residues) and PRD (~20 residues). The KD is intimately associated with the FAT domain, which consists of TRD1, TRD2, TRD3, and HRD subdomains. The conformation of the Tra1 FAT domain is largely similar to that of other PIKK members. The kinase N-lobe insertion known as FRB (FKBP12-rapamycin-binding) shows substantial conformation variation. In the mTOR and DNA-PKcs, the FRB domain closely contacts the N-lobe, along with the LBE on the opposite site of the cleft, restricting the active site[33,50]. However, the Tra1 FRB moves away from the N-lobe and directly packs against the LBE domain, which encloses the Tra1 active site (Fig. 3c and Supplementary Fig. 7). The conformation difference of the Tra1 FRB could exert a substantial effect on the architecture of the pseudokinase domain.

**Architecture of the Tra1 pseudokinase domain**. Surprisingly, the Tra1 pseudokinase domain has the signatures of an active conformation, which harbors fully ordered structural elements crucial for catalysis, including activation loop, catalytic loop, and P-loop (Fig. 3d). There are some structural differences compared with the other PIKK members. Especially, Tra1 harbors a much larger activation loop ($^{3585}$EMLPSRFPYERVKPLLKNHDLS LPPDSPIFHNNEPVPFR$^{3623}$), containing a 17 residues insertion (underlined residues), which adopts a fully extended conformation that stabilize the active site. In the DNA-PKcs[33], mTOR[50], and ATR/Mec1 kinases[36], the PRD domain (PIKK regulatory domain) is almost paralleled to the kα9 and the activation loop is enclosed by the PRD and FATC domains (Supplementary Fig. 8).

While the Tra1 FATC remains in its place, the PRD domain moves outwards by 45° relative to the kα9 (Fig. 3d). In this way, the Tra1 PRD makes way for the activation loop and therefore substantially increases the exposure of the activation loop. It has been proposed that the PRD domain regulates the KD of PIKKs through interactions with the activation loop[51]. The synergistic conformational differences of the PRD and activation loop of Tra1 likely reflect the divergence of its non-catalytic features. Like other active PIKK kinases, Tra1 harbors an active kinase conformation and single residue mutations in pseudokinase domain substantially affect Tra1 function[52]. These structural and functional observations consistently suggest that the active conformation of the pseudokinase domain is important for the non-catalytic functionalities of Tra1.

**Assembly of the actin/Arp module**. Eaf1 subunit is an assembly scaffold of the NuA4 complex, making distinct physical interactions with all four different functional modules (Fig. 4a)[21]. Consistently, the Eaf1 is corresponding to the Neck density in the TEEAA assembly, which bridges different modules (Fig. 4b). Act1 and Arp4 are shared by NuA4 with the SWR1 and INO80 chromatin remodeling complexes. The Act1 and Arp4 subunits reside at the peripheral Head region, which are assembled into the NuA4 complex through binding to Eaf1 HSA domain (Fig. 4b and Supplementary Fig. 9). Similar to the SWR1 HSA-Arp4-Act1 assembly[46], the Eaf1 HSA binds to the Arp4 and Act1 through hydrophobic interactions (Supplementary Fig. 10a). Three key hydrophobic residues of Eaf1 HSA (L362, M369, and F373) make multiple hydrophobic interactions with the M1 and L462 residues of the Arp4 (Fig. 4c). The other three key residues of Eaf1 HSA (I383, M387, and I391) contacts multiple hydrophobic residues in Act1, including F169, F375, M355, F352, I345, and L349 (Fig. 4d). Recruitment of NuA4 to DNA damage sites is mediated in part by the Arp4 subunit[53]. The actin *act1-136* mutant (with mutation

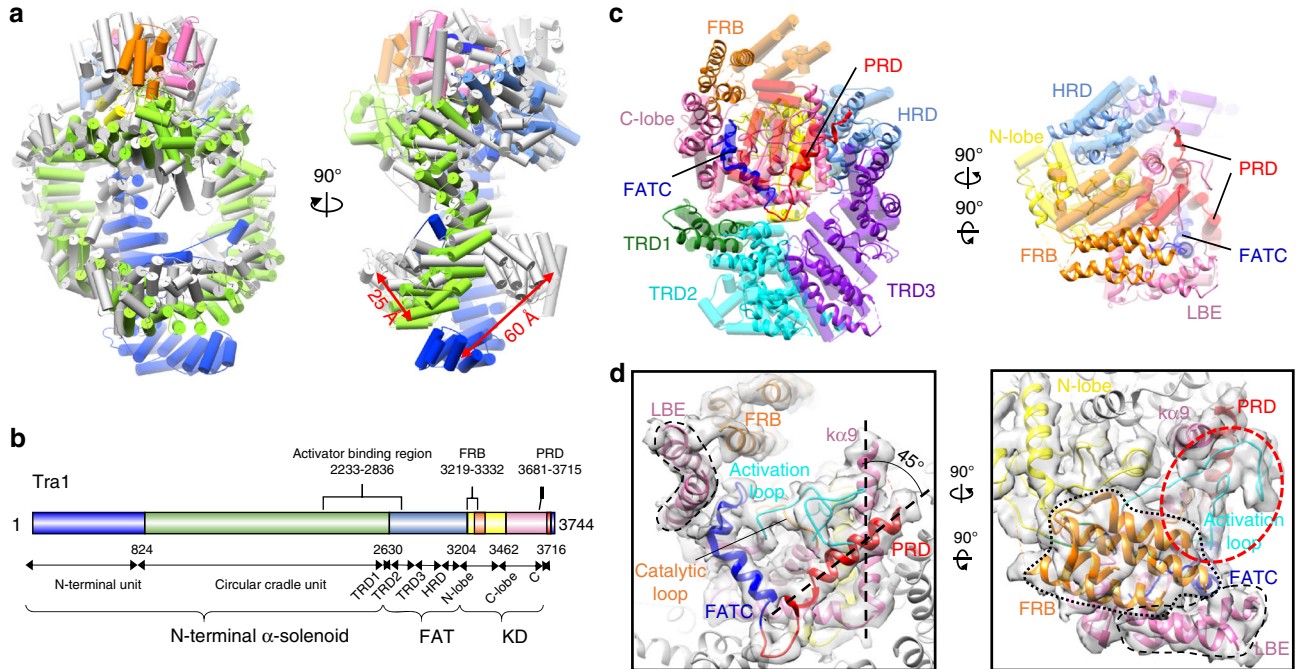

**Fig. 3** Structure of the Tra1 in the TEEAA assembly. **a** Structure comparison of Tra1 with DNA-PKcs. The Tra1 subunit is color-coded by domain assignment as panel **b** and aligned with the crystal structure of DNA-PKcs (gray, PDB ID: 5LUQ). Significant differences in the two structures are observed in the N-terminal HEAT repeats. The N-terminal unit of Tra1 moves inwards by 60 Å, while the circular cradle unit moves a smaller distance by 25 Å. **b** A schematic representation highlighting the functional domains of Tra1. Three units of Tra1: N-terminal HEAT repeat (the N-terminal unit and the circular cradle unit), the FAT (TRD1, 2, 3 and HRD), and the kinase regions are labeled. The activator binding region, the FRB insertion and the PRD domain are labeled above the schematic. **c** Structural comparison of the FATKIN region of Tra1 and DNA-PKcs. Tra1 FATKIN is shown in ribbon diagram model and DNA-PKcs FATKIN is shown in transparent pipes. The domain organization and color assignment are the same as Supplementary Fig. 7. **d** Two detailed views of the active site of Tra1, with the N-lobe is shown in yellow, the C-lobe in hot pink, the FRB insertion in orange, PRD in red, and FATC in blue. The cryo-EM density is displayed in transparent surface. The activation loop is colored in cyan and the catalytic loop in orange. The FAT domain is shown in gray. The LBE domain is highlighted in black dashed lines, while the FRB insertion in dotted lines. PRD moves outwards by 45° relative to the kα9. The red dashed circle highlights the highly extended and exposed activation loop

D2A in subdomain 1) affects the global acetylation of histone H4[54]. The actin D2A site is exposed, which suggests a potential interaction surface of the NuA4 complex with its substrate (Fig. 4b).

**Interaction of Eaf1 with Tra1.** The Eaf1 SANT domain directly binds to the Tra1 subunit (Fig. 4e, Supplementary Fig. 10b and 11)[6], which is mediated primarily through the intermolecular hydrogen bonds and ionic bonds. Especially, the charged residues in the SANT domain (R691, R690, and E698) are in the close proximity to the negative residues in the Tra1 LBE (D3605, E3512) and a positive residue (R3731) in the FATC (Fig. 4e). Besides the LBE and FATC domains, the Eaf1 subunit also packs against to the TRD1 region of the Tra1 FAT domain (Figs. 1d and 4b). These multiple interactions stabilize Eaf1 binding to the gigantic Tra1 subunit, which serves as another scaffold for NuA4 and SAGA complexes[40]. Two mutations within the FATC domain (*tra1-L3733A* and *tra1-F3744A*) results in slow growth under stress conditions[55]. The phenotypes carrying mutations in the FATC domain are likely due to destabilization of the Eaf1–Tra1 interactions.

**Interaction of Eaf5 with Eaf1 and Tra1.** The Eaf1 subunit is involved into interactions with Eaf5, which anchors the dissociable Eaf5/7/3 to the NuA4 complex[6,9]. The N-terminal 200 residues of Eaf1 forms a generally disordered loop region. In contrast, Eaf5 is composed of largely ordered α-helices (Fig. 4f and Supplementary Fig. 12). Besides its close proximity to the N-terminal of the Eaf1, the Eaf5 subunit also makes extensive contacts with the Tra1 FAT domain (TRD1 and TRD2 regions)

and the Circular Cradle Unit (45R–51R HEAT repeats). The arrangement of only helices at the intermolecular interface generates a large curved surface (Fig. 4f). Eaf5's loose binding to Eaf1 and its extensive interactions with Tra1 offers the elasticity to the Eaf5/7/3 module to be dynamically assembled into NuA4 complex. Interestingly, the Tra1 TRD1-TRD2, and HEAT 45R–51R interfaces binding Eaf5 are also involved in interaction with transcription factors[39]. It is tempting to speculate that the dissociable Eaf5/7/3 module and transcription factors could competitively interact with the Tra1 subunit to modulate the recruitment and activity of the NuA4 complex.

**Assembly of the piccolo module.** The piccolo module, responsible for maintaining global acetylation of chromatin[17,18], can be dynamically assembled into the NuA4 complex (Fig. 5a). After 3D classification of NuA4 particles (Supplementary Fig. 3), a subset of particles corresponding to the TEEAA-piccolo assembly was subjected to high-resolution refinement, resulting in a 3D density map with a final overall resolution of 7.6 Å (Fig. 5b and Supplementary Fig. 13). The piccolo module harbors prominent structural features of a four long helices bundle[56]. Subsequent unambiguously docking of the piccolo crystal structure (PDB ID: 5J9U)[56] into the cryo-EM map resulted in high correlation coefficients (Supplementary Fig. 14). Compared with the NuA4 TEEAA assembly, the Eaf1 conformation is much more contractive after association with piccolo (Fig. 5b).

As the scaffold for NuA4 complex assembly, the structural plasticity of Eaf1 reflects the need to adapt to interact with

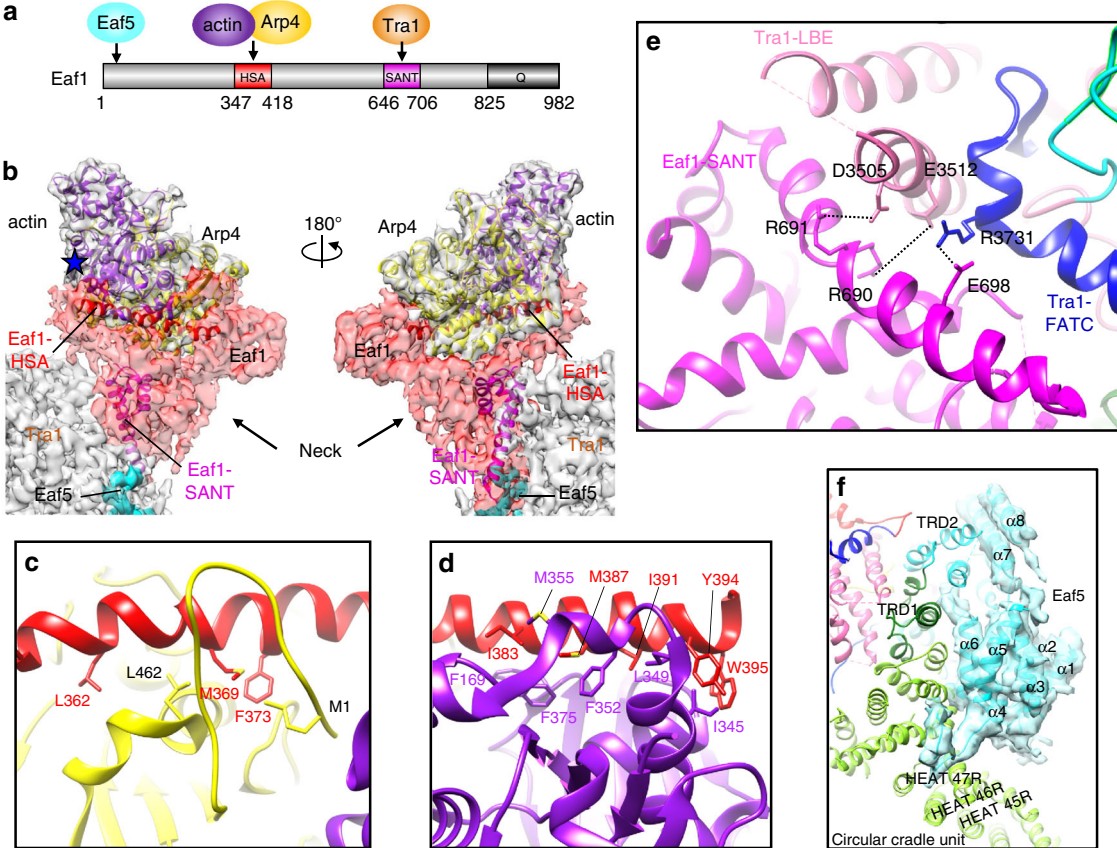

**Fig. 4** Interaction between TEEAA subunits. **a** Domain architecture of Eaf1 is colored in red (HSA domain), magenta (SANT domain), dark gray (C-terminal rich-Q domain), and gray (other regions). **b** The cryo-EM density is shown as a translucent surface fitted with the ribbon diagram model of actin, Arp4, Eaf1 HSA domain, and SANT domain. The Eaf1 density is shown as red translucent surface and the Eaf5 is shown in cyan solid surface. Both actin and Arp4 subunits directly interact with the HSA domain of Eaf1. The actin D2A mutation (*act1-136* mutant) is highlighted with a blue star. **c** HSA^Eaf1-Arp4 interaction. Three key hydrophobic residues of HSA^Eaf1 (L362, M369, and F373) contact the hydrophobic M1 and L462 residues of Arp4. **d** HSA^Eaf1-actin interaction. Three key residues of HSA^Eaf1 (I383, M387, and I391) are labeled, which contacts multiple hydrophobic residues in actin, including F169, F375, M355, F352, I345, and L349. **e** Interaction between Tra1 and Eaf1 subunits. The intermolecular hydrogen bonds between the SANT domain of Eaf1 and FATC and LBE domains of Tra1 are denoted with black dashed lines. **f** Interaction between Tra1 and Eaf5 subunits

different subunits and cofactors to mediate different physiological functions in the cell. The putative conformational changes of Eaf1 could facilitate delivering the nucleosome substrate binding to the actin/Arp4 module at the Head to the catalytic piccolo module. Consistently, the SAGA complex also harbors a continuous twisting movement with respect to Tra1, which has been proposed to be an important feature of SAGA to facilitate the search for a substrate nucleosome[45].

Epl1 C-terminus interacts with Eaf1 through the HSA domain and a highly charged region (aa 329 to 538)[6] and Yng2 also binds to Tra1[57], bridging the piccolo module to the rest of NuA4. Consistent with these biochemical and functional findings, the C-terminal of Epl1 points to the Eaf1 HSA domain and Yng2 packs towards the Circular Cradle unit of the Tra1 (Fig. 5c, d). In addition, Esa1 is in close contact with the HEAT repeat 41R and the N-terminal of Epl1 packs against the HEAT repeat 24R (Fig. 5d). Consistently, the Δ24R or Δ41R mutant substantially reduces the stability of the SAGA and NuA4 HAT modules (Supplementary Fig. 15a)[39]. These structural and functional observations consistently highlight the Tra1 subunit, especially the Circular Cradle unit, as the scaffold for HAT module assembly.

**Validation of the NuA4 subunit interfaces**. Our model of NuA4 architecture provides a wealth of testable structure predictions.

We focused on the scaffolding Eaf1 subunit to perform in vitro and in vivo experiments to test specific interaction surfaces. Using GST pull-down assays, we confirmed that purified Eaf5/7/3 (TINTIN) module and recombinant Tra1 C-terminus (FRB-PI3K-FATC) bind Eaf1 N- and C-terminal halves, respectively (Fig. 6a, b). We then produced different deletions in vivo within the Eaf1 scaffold subunit[6]. This allowed us to determine that TINTIN associates within the first 250 amino acids of Eaf1 (Fig. 6c, d). It also showed that Eaf1 HSA domain (aa346–420) is critical for the association piccolo NuA4 and SWR1-C modules but not Tra1 or TINTIN. These results along with our cryo-EM-based model and previous work[9] indicate that Eaf5 anchors TINTIN to NuA4 through interaction with Eaf1 N-terminus. Phenotypic analysis of the yeast cells carrying Eaf1 deletions supports the detected loss of subunits/modules but also suggest that some additional sequences are required for proper NuA4 tri-dimensional assembly and function, in agreement with our model showing the multiple interfaces with the other scaffolding subunit Tra1 (Fig. 6e and Supplementary Table 2).

We then mutated, predicted key hydrophobic residues within Eaf1 HSA domain. As predicted, the triple mutant completely loses interaction with the SWR1-C shared module (Arp4-Act1-Swc4-Yaf9) in vivo (Fig. 6f). It also cripples interaction with the piccolo NuA4 module but retains some association. These results suggest that piccolo NuA4 can interact with Eaf1 independently

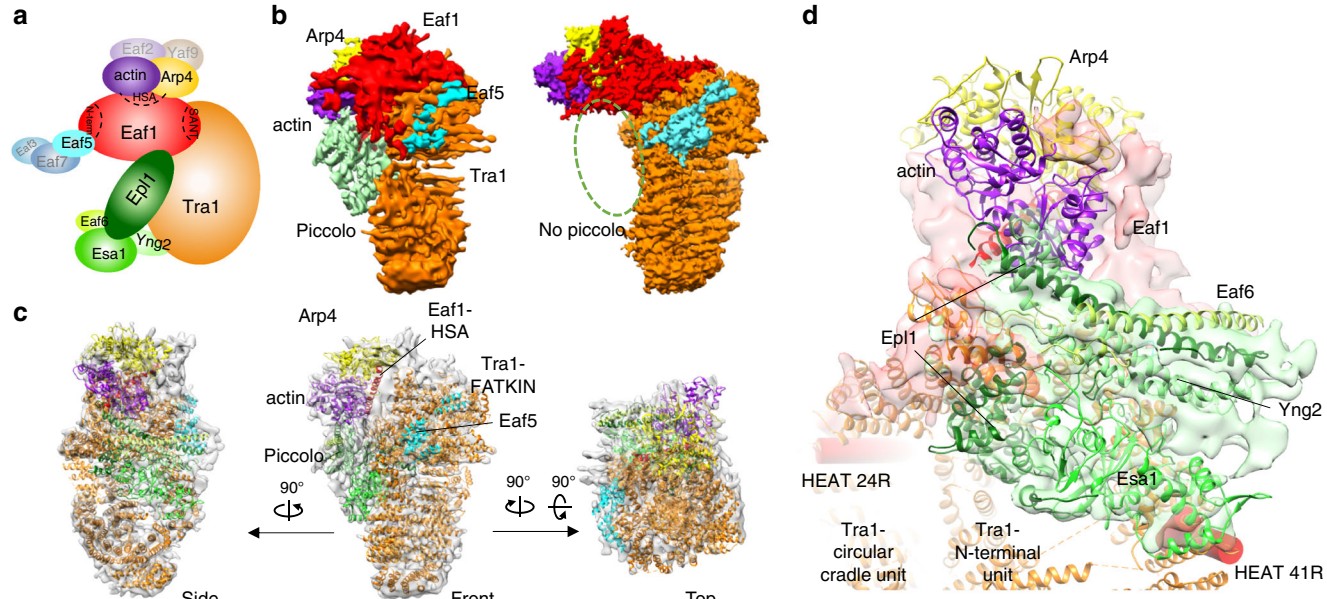

**Fig. 5** Architecture of the NuA4 TEEAA-piccolo assembly. **a** Schematic view of subunit and modular organization of the NuA4 complex, highlighting the subunits present in the TEEAA-piccolo assembly. **b** Comparison of the cryo-EM structures of the TEEAA-piccolo and TEEAA assemblies in the front view. The density map is color coded by subunit assignment as panel **a**. The green dashed oval indicates the binding position of the piccolo module. **c** Three views of the TEEAA-piccolo assembly. The cryo-EM density is shown as a translucent surface and fitted with the ribbon diagram model of the piccolo crystal structure (PDB ID 5J9U). Arp4 is in yellow, actin in purple, Tra1 in orange, HSA domain of Eaf1 in red, Eaf5 in cyan, and the piccolo module in green. Each successive view is rotated as indicated. **d** Enlarged view of the piccolo assembled into the NuA4 complex. Epl1 is in dark-green, Esa1 in lime, Eaf6 in yellow-green, and Yng2 in light-green. Piccolo interacts with both Eaf1 and Tra1 subunits. The Tra1 HEAT 24R and 41R mainly responsible for piccolo binding are highlighted in red pipes

of the Arp4/Act1 module but is greatly stabilized by it, as suggested by the close proximity of Epl1 C-terminus, Eaf1 HSA domain and Arp4 in our model.

Finally, we investigated Tra1 association with NuA4/Eaf1 in vivo. Deleting Eaf1 SANT domain or mutating the predicted key charged residues has only small effect on Tra1 association with the complex (Fig. 6g and Supplementary Table 2). This is likely explained again by the fact that the very large Tra1 protein shows multiple distinct interactions within the NuA4 complex, as depicted in our 3D model. Nevertheless, the loss of known NuA4-interacting Yap1 transcription factor[58,59] co-purifying with the NuA4 complex mutated in Eaf1 SANT domain suggests a defect in Tra1 recruitment function (Supplementary Table 2).

## Discussion
The assembly of SAGA and NuA4 shares the same Tra1 surfaces (Supplementary Fig. 15a)[39]. The SAGA assembly interface is centered on the FAT domain and the upper half of the N-terminal HEAT repeats of Tra1 (Supplementary Fig. 15b)[60]. Besides the shared interface on Tra1, the segments of Tra1 LBE/FATC domains are also involved in NuA4 assembly (Supplementary Fig. 16). Each PIKK member generally works as part of larger assemblies with several accessory and regulatory subunits. Interestingly, Tra1 and mTOR use the similar surface to assemble the NuA4 and mTORC1, especially the N-terminal HEAT repeats and LBE domain (Supplementary Fig. 16)[61,62]. The interaction surfaces of Tra1 with VP16 activator are dispersed across the Circular Cradle, N-terminal unit, and Head regions (Fig. 7a)[63]. The prominent VP16 binding region corresponds to TRD1-TRD2 and HEAT 42R-51R of Tra1 (residue 2233–2836), which is also the Eaf5 binding interface to bridge the dissociable Eaf5/7/3-TINTIN module (Figs. 7a and 4f). This supports again the idea that activators and Eaf5 may compete for Tra1 binding to regulate NuA4 recruitment and function.

TRRAP (human Tra1) plays a critical role in maintaining a tumorigenic, stem cell-like state[64]. The HEAT repeats region and FAT domain are hotspots for oncogenic mutations, however, the effects of such mutations on TRRAP function are unclear[65,66]. Unexpectedly, the TRRAP mutations found in human cancers are largely centered on the Tra1 interaction surfaces mediating assembly of NuA4 and SAGA complexes, which also harbor several post-translational modifications (UniProtKB—Q9Y4A5) (Fig. 7b and Supplementary Fig. 15). Such oncogenic mutations on TRRAP thus likely result in destabilizing the assembly and functions of the HAT complexes. These considerations warrant targeting the scaffolding function of TRRAP with small-molecule modulators, which may prove successful in preventing cancer and/or in killing cancer cells.

Tra1/TRRAP has evolved as a specialized PIKK member to serve as a scaffold for multiprotein assemblies or as a platform for recruitment of different regulatory factors to chromatin[40]. Although Tra1 lacks kinase activity, it adopts kinase active conformation in the NuA4 complex. The Eaf1 SANT domain tightly binds to Tra1 LBE and FATC domains, which serve as scaffold for NuA4 complex assembly. The actin/Arp4 module mainly interacts with Eaf1, while the Eaf5/7/3 and piccolo modules largely pack against the Tra1. Cancer-linked mutations found in TRRAP are clustered in the NuA4/TIP60 assembly interface, which are also hotpots for post-translational modifications. Our study not only elucidates the detailed architecture and molecular interactions within the essential NuA4 complex but also provides exciting insights into the scaffolding and regulatory mechanisms of the Tra1/TRRAP pseudokinase.

## Methods
**Purification of yeast NuA4.** *S. cerevisiae* NuA4-FLAG (*MATa ESA1-FLAG his3Δ200 leu2Δ0 met15Δ0 trp1Δ63 ura3Δ0*) strain was grown in YPD medium to the stationary phase. About 100 g cells were harvested, washed, and re-suspended in extraction buffer (50 mM HEPES [pH 7.6], 300 mM KOAc, 0.5 mM EDTA, 5

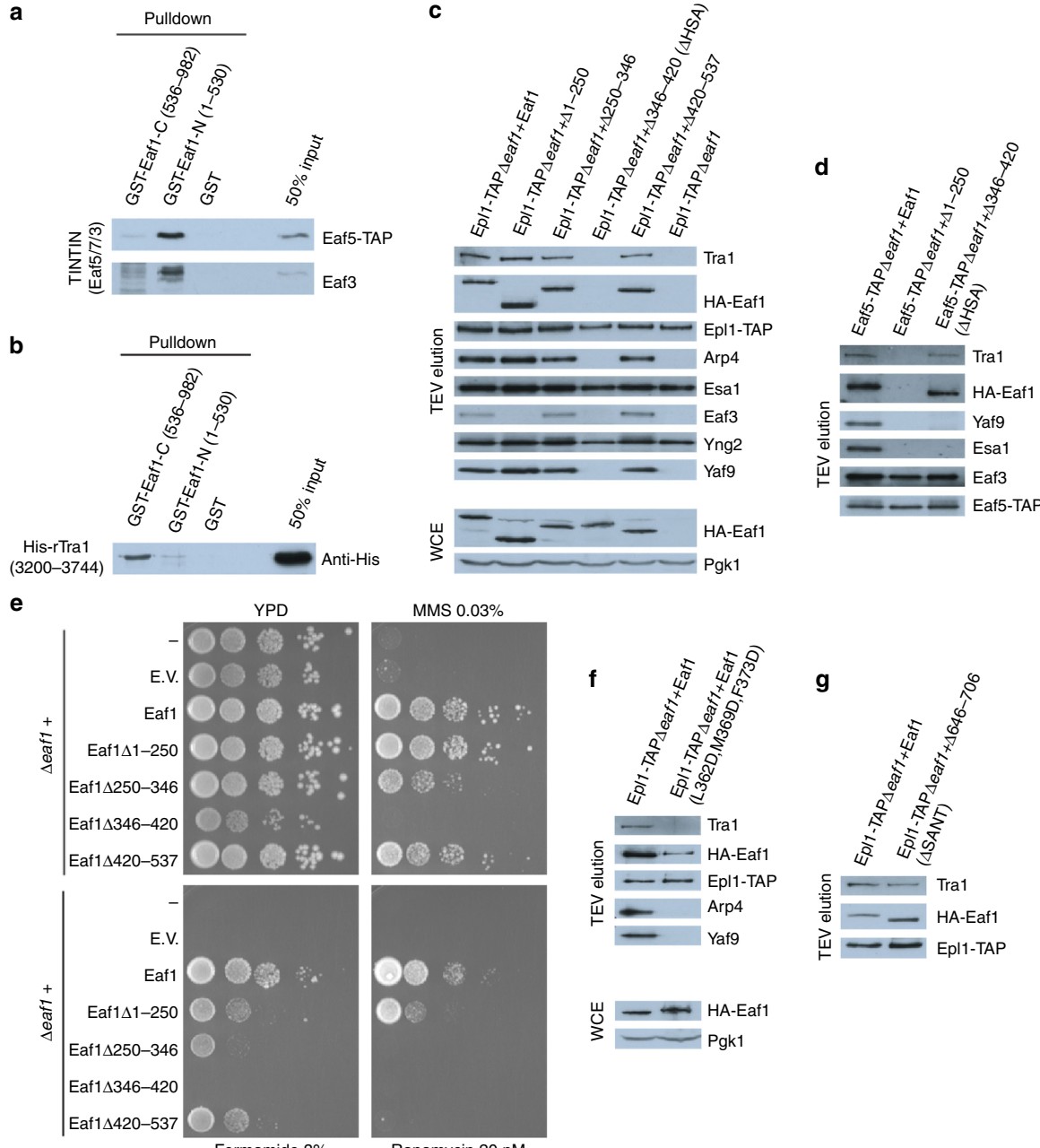

**Fig. 6** Functional interactions between NuA4 subunits. **a** The TINTIN subcomplex (Eaf5-Eaf7-Eaf3) interacts with the N-terminal portion of Eaf1 in vitro. Purified native TINTIN was incubated with recombinant GST-tagged Eaf1 (1–530), Eaf1 (536–982), and GST beads followed by washes and western blotting. **b** Tra1 interacts with the C-terminal region of Eaf1 in vitro. Recombinant His-tagged Tra1 (3200–3744) was incubated with GST fusion proteins on beads as in **a**. **c**, **d** Dissection of Eaf1 interaction surfaces in vivo. NuA4 was purified through Epl1 (**c**) or Eaf5 (**d**) subunits, from cells expressing wild type or truncated forms of HA-tagged Eaf1 (amino acid positions indicated) and analyzed by western blotting using the indicated antibodies. Expression levels of the different Eaf1 truncation mutants are similar as shown in whole cell extracts (Pgk1 signal is for loading control). The piccolo NuA4 module (Epl1-Esa1-Yng2-Eaf6) and the shared SWR1-C module (Arp4-Act1-Swc4-Yaf9) require the HSA domain of Eaf1, while TINTIN binds within its first 250 amino acids. Tra1 still associates with Eaf1 in the different truncations analyzed. **e** Cells expressing Eaf1 truncation mutants show varying growth defects when challenged with drugs/chemical reagents. Tenfold serial dilutions of yeast cultures were spotted on plates that were supplemented with MMS, Formamide or Rapamycin. ("−" indicates Δeaf1 strain without the plasmid vector; E.V., empty vector). Loss of TINTIN from NuA4 leads to sensitivity to rapamycin and formamide while loss of Piccolo and SWR1-C modules leads to growth defect in rich media and are non-viable in presence of DNA damage (MMS), formamide and rapamycin. Eaf1 mutants that associates with the full set of NuA4 subunits are also showing growth defects, indicating a partly non-functional NuA4 complex. **f** Loss of SWR1-C module with specific point mutants in Eaf1-HSA domain. Arp4 is required for efficient binding of piccolo NuA4 module to Eaf1 in vivo (experiment as in **c**). **g** Deletion of Eaf1-SANT domain results in decrease interaction between Tra1 and Eaf1 in vivo (experiment as in **c**). The decrease in Tra1 signal is apparent after correction with Epl1 and Eaf1 signals in the fraction and mass spec data (Supplementary Table 2). Complete western blots can be found in Supplementary Figs. 17, 18

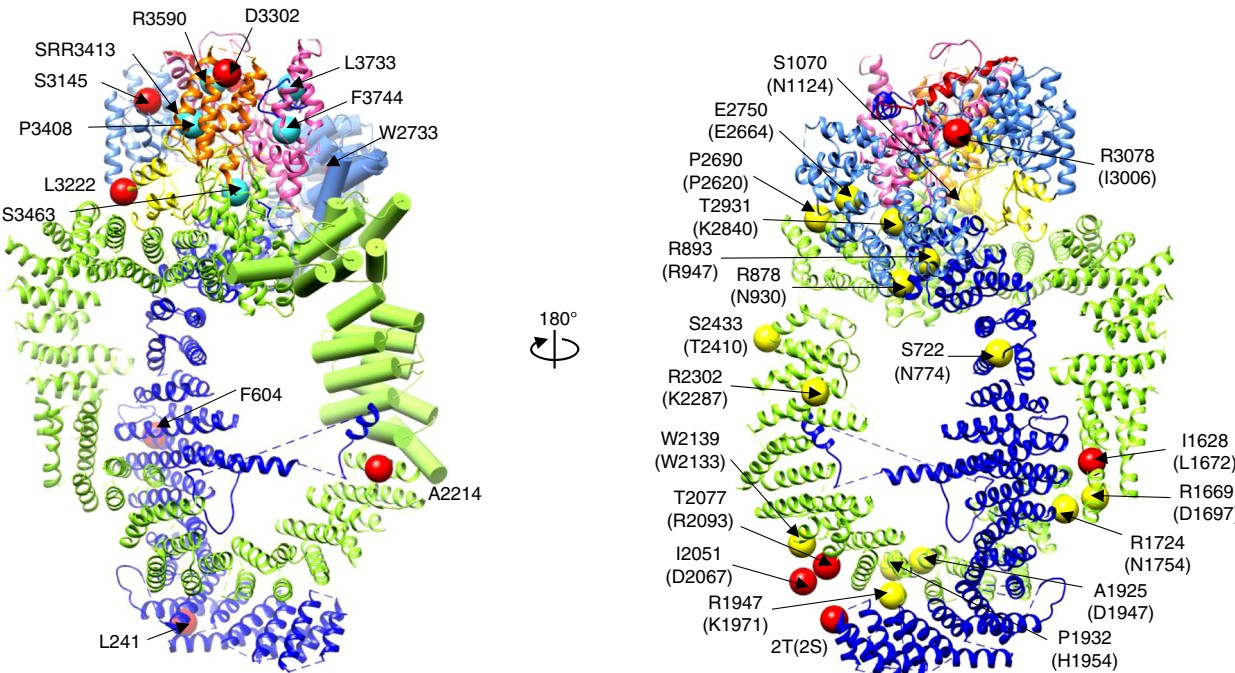

**Fig. 7** Scaffolding function of the Tra1/TRRAP. **a** Mutations disrupting the interaction of Tra1 with VP16 activator are mapped to the structure elements as red spheres. The Tra1 mutations causing growth defects are shown as large cyan spheres. The prominent VP16 binding region corresponding to TRD1–TRD2 and HEAT 42R-51R of the Tra1 (residue 2233–2836) is highlighted in pipes. **b** The TRRAP mutations in human cancer are shown as yellow spheres and modification sites (acetylation and phosphorylation) reported for TRRAP are shown as red spheres with structural elements colored as in Fig. 3b

mM β-ME, 10% (v/v) glycerol, 0.1% (v/v) NP-40 and protease inhibitors) and a whole-cell extraction (WCE) was prepared[67]. This WCE was selectively pre-cipitated in 30–55% ammonium sulfate and re-suspended using 1X TEZ buffer (50 mM HEPES [pH 7.6], 1 mM EDTA, 10 μM ZnCl₂, 10% (v/v) glycerol and protease inhibitors). The supernatant after centrifugation was incubated for 2 h at 4 °C with 1 ml of a 50% slurry of FLAG resin beads (Sigma) that had been pre-equilibrated with 1X TEZ plus 250 mM ammonium sulfate. Then, the beads were washed with 50 ml of 1X TEZ plus 500 mM ammonium sulfate, followed by a second wash with 50 ml of 1X TEZ plus 50 mM ammonium sulfate. Afterwards, 1X TEZ plus 100 mM ammonium sulfate, 10 mM 3X FLAG peptide (Sigma) was added to the resin beads and incubated overnight at 4 °C. The NuA4 fraction was eluted with three column volumes of 1X TEZ plus 100 mM ammonium sulfate, which was snap-frozen in liquid nitrogen and temporarily stored at –80 °C. For the next purification step, the FLAG elute fractions were thawed on ice and applied onto a Mono Q column (GE Healthcare) in Q100 buffer (1X TEZ plus 100 mM ammonium sulfate, 0.02% NP-40, 10 mM β-ME, 10% (v/v) glycerol) and was resolved over a 100–1000 mM ammonium sulfate gradient. The NuA4 complex elution was flash-frozen in liquid nitrogen, analyzed by SDS-PAGE and EM examination.

**Negative stain EM analysis**. The Mono Q peak fraction was diluted 8–12 times (20 mM HEPES [pH 8.0], 40 mM KOAc, 5 mM MgCl₂, 0.1% trehalose, 2 mM DTT, 0.01% NP-40) and applied to a carbon-coated 400-mesh Cu EM specimen grid freshly glow discharged and was then preserved by staining with 0.75% (w/w) volume uranyl formate solution. Images were recorded on Tecnai F20 electron microscope (FEI) operated at 200 kV and 4096 × 4096 CCD detector (FEI Eagle) at a magni-fication of ×62,000 with a defocus range of 0.6–0.8 μm, resulting in a calibrated sampling of 3.54 Å per pixel after two-fold pixel binning of the original CCD images.

3D reconstructions under negative stain were calculated by the random conical tilt method (RCT)[68]. Tilted (-55°) and un-tilted image pairs were obtained under low-dose conditions and particles were selected using the TiltPicker program[69] and montaged them for interactive screening, yielding ~22,500 particles of the NuA4 complex. Iterative alternating rounds of supervised multi-reference alignment and classification as well as reference-free alignment were run to improve the homogeneity of the image classes with SPIDER[70] and SPARX[71].

**Sample vitrification and Cryo-EM data collection**. The NuA4 complex was diluted to final concentration 20–50 μg/ml (20 mM HEPES [pH 8.0], 40 mM KOAc, 5 mM MgCl₂, 0.1% trehalose, 2 mM DTT, 0.01% NP-40) and 3 μl of ali-quots were applied to freshly glow discharged Quantifoil R2/1 grids coated with a second layer of thin carbon film. The grids were blotted for 3–4 s at 4 °C in 100% humility, then plunged into liquid ethane using an FEI Vitrobot (FEI Company).

Frozen grids were stored in liquid nitrogen. A batch of grids were prepared under the same condition and one or two grids were used to check the quality of the sample vitrification on FEI Tecnai TF20 electron microscope. The grid would be recovered if the vitrification looked good. Then, the recovered grid along with the same batch of grids would be loaded into a Titan Krios microscope for data collection. High resolution images were recorded on Titan Krios equipped with a field emission source operated at 300 kV and K2 direct electron detector at a nominal magnification of 18,000 with a defocus range of 2.0–3.0 μm, resulting in a calibrated sampling of 1.3 Å per pixel. The total accumulated dose rate was set to be 30 e² per Å² on the specimen and the exposure time was 7.5 s. Each image was fractionated into 30 frames.

**Image processing**. Frames were summed to a single micrograph for subsequent processing using motion correction procedure written by Xueming Li[72]. CTF parameters and defocus values for each micrograph were determined by using CTFFIND3[73]. A semi-automated procedure using RELION[74] and Gautomatch[75] was used to pick particles. Totally, 143,084 NuA4 particles were picked after 2D classification and manual cleaning. All the 3D classification and auto refinement was performed using RELION. The initial RCT model of NuA4 state II was low-pass filtered to 60 Å and used as the starting model for the 3D classification. A final set of particles corresponding to the NuA4 state II (class 6, 19,060 particles) was subjected to 3D refinement without symmetry. The rest 124,024 particles were subjected to the second round 3D classification using the resulting class 2 volume as a reference model. Two out the 6 classes (class 4 and 5) were merged to obtain a set of 63,197 particles. Then, the selected two subsets with 63,197 and 19,060 particles were subjected to "auto-refinement" and "postprocessing" procedures, respectively. The two resulting reconstructions were calculated at resolutions of 4.68 Å and 7.64 Å based on the gold-standard fourier shell correlation (FSC) 0.143 criterion. Local resolution maps were calculated using ResMap[76]. All the 3D structures were displayed by Chimera[77].

**Model building and simulation**. For Tra1 subunit, a model of the FAT-KD-FATC domain (1114 aa) was generated based on the crystal structure of DNA-PKcs (PDBID: 5LUQ)[33] using SWISS MODEL[78]. Rigid body refinement was then per-formed using Jigglefit in coot[79]. The model of the FAT-KD-FATC domain well matched with the densities of the cryo-EM map and some mismatched regions were then manually adjusted in coot according to the EM density. De novo model building of the region other than FAT-KD-FATC domain of Tra1 protein was performed using COOT by taking into account of the secondary structural pre-dication. The crystal structure of actin ternary complex (PDBID: 5I9E)[46] con-taining Arp4, actin, and the HSA domain of Swr1 could be unambiguously docked into the EM density map. The sequence assignment of Eaf1 HSA domain was

substituted in COOT. The SANT domain of Eaf1 was homology modeled based on the NMR structure of SANT domain of human DNAJC2 (PDBID: 2M2E). The Eaf5 structure is de novo modeled based on the secondary structure prediction and the remaining EM density. Real-space refinement (phenix.real_space_refine) in Phenix[80] was used for model refinement.

**Yeast strains and reagents**. Plasmids, yeast strains, and growth assays were prepared following standard procedures[6,9]. Expression of mutant Eaf1 proteins in vivo was based on an ARS/CEN low copy vector with a native *EAF1* promoter, covering an *eaf1* deletion background. Primer sequences used for cloning can be found in Supplementary Table 3.

**Tandem affinity purification**. Tandem affinity purification (TAP) was performed as described previously[9]. In brief, 250 ml of yeast culture was lysed using glass beads in extraction buffer (20 mM HEPES pH 7.4, 300 mM NaCl, 0.1% NP-40, 2 mM $MgCl_2$, 5% glycerol, 1 mM DTT, 1 mM PMSF, protease inhibitors). The lysate was incubated with IgG (Millipore) coupled Dynabeads (M-270 Epoxy, Invitrogen) for 4 h at 4 °C followed by washing and elution with TeV protease. The elution was incubated with Protein A sepharose beads to remove leaked IgG. Antibodies used for western blotting were anti-Yaf9 (Abcam 4468), anti-Esa1 (Abcam 4466), and anti-HA(Roche). Anti-Tra1, anti-Yng2, anti-Eaf3, and anti-Arp4 were kindly provided by P. Grant, S. Tan, J.C. Lucchesi, and D.J. Stillman, respectively.

**Pull down assay**. Recombinant proteins were purified using standard protocol[17]. The bacterial culture was lysed using lysozyme followed by sonication. The cell lysate was incubated with glutathione sepharose (GE Healthcare) or Ni-NTA bead (QIAGEN) for 4 h at 4 °C. His-tagged protein was eluted using imidazole whereas the GST-tagged proteins were stored at 4 °C immobilized on the glutathione sepharose beads. The pull down assays were performed using 300–600 ng of immobilized GST-tagged proteins incubated with equivalent amount of His-tagged proteins for 3–4 h at 4 °C (1 μg of GST protein was used as control). The beads were washed with binding buffer (25 mM HEPES pH 7.5, 100 mM NaCl, 10% glycerol, 100 μg/ml BSA, 1 mM PMSF, 0.5 mM DTT, 0.1% Tween-20, protease inhibitors) and ran on SDS-PAGE followed by western blotting with anti-TAP (Open Bio-system CAB1001), anti-Eaf3 (Abcam 4467), and anti-His (Clontech) antibodies.

**Data availability**. The accession number for the cryo-EM density maps reported in this paper are EMD-6815 and EMD-6816. The accession number for the structural coordinates of the NuA4 TEEAA assembly is PDB: 5Y81. Other data supporting the findings of the manuscript are available from the corresponding authors upon reasonable request.

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

## Acknowledgements

The Cryo-EM data collection was supported by the National Center for Protein Science (Shanghai, China). We thank Prof. Xuetong Shen (University of Texas MD Anderson Cancer Center) for providing the ESA1-FLAG tagged *S. cerevisiae* strain: NuA4-FLAG (*MATa ESA1-FLAG his3Δ200 leu2Δ0 met15Δ0 trp1Δ63 ura3Δ0*). We thank Mr. Ruifeng Chen for initiating the research project and thank P. Tang for her support with EM facility at the USTC. We thank Anne-Lise Steunou for providing the purified TINTIN complex used in pulldown assay. We thank Tengwei Wu for characterization of several NuA4 deletion mutants. This work was supported by grants from the National Basic Research Program (2014CB910700) and the National Natural Science Foundation of China (31170694) to G.C., the National Natural Science Foundation of China (31570726) to X.W., and the Canadian Institutes of Health Research (FDN-143314) to J.C. J.C. holds the Canada Research Chair on Chromatin Biology and Molecular Epigenetics.

## Author contributions

G.C., X.W., and J.C. designed research; X.W., S.A., and Z.Z. performed research; G.C., X. W., and J.C. analyzed data and wrote the paper.

## Additional information

**Competing interests:** The authors declare no competing interests.

