## [Peer Review File(PDF 823 kb) · Nature Communications]

Reviewers' comments:

Reviewer #1 (Remarks to the Author):

The structural characterisation of NuA4 is required to fully understand the mechanisms of this critical cell regulator. In their manuscript Wang et al describe cryo-EM structures of two NuA4 assemblies. These are relevant results, but there are some issues needing some attention.

Page 5, lines 9-12: "past structural analyses have been limited to low resolution structures that suffer from deformation artifacts induced by chemically crosslinking and/or by staining with heavy metals". The authors may consider revising this statement. While the previous structures are at lower resolutions, is there any direct evidence they are distorted as suggested?

Page 6, lines 6-7: "Thus, we obtained highly homogeneous NuA4 complex suitable for cryo-EM analysis". The authors need to revise this statement. It is a fact that they obtained a sample suitable for cryo-EM, but not a highly homogeneous one. The image analysis revealed compositional heterogeneity in the sample with the presence of different sub-complexes and in fact no structure with all 13 subunits was obtained.

Page 6, lines 12-18. The authors did a random conical tilt (RCT) analysis of negatively stained sample in order to obtain a starting model for the analysis of the cryo-EM data, and indeed they obtained classes that proved suitable for this purpose. They describe the two dominant classes obtain from RCT as two different assembly states, based on a visual comparison. While this can be the case, the authors also need to discuss the possibility that the differences between the two classes can be the result of map distortions due to the missing cone and the flattening effect observed when RCT is done using images from negative stain. While the resulting maps may suffer from such distortions, they may still sufficiently accurate to make them suitable starting models for refinement.

Page 6, lines 18-19: "Both maps reveal a diamond ring shaped density at the base". For clarity, this density should be labelled in the maps shown in Extended Data Fig 1c.

Page 6, line 29: "resulting in a 3D density map with final overall resolution of 4.7 Å". At this resolution secondary structure elements should be clearly resolved. Unfortunately in the current version of the manuscript there are no figures to unambiguously show that this is the case. This can be solved by adding new panels with close-up views of the map clearly showing the secondary structure. Improving the figures where the maps are shown with fitted coordinates can also help to evaluate the level of detail resolved in the maps, since in the current version of the manuscript in all of such representations the maps are rendered as almost invisible transparent surfaces that do not allow appreciating how well the coordinates really fit. Finally, the labels for each of the FSC curves shown in Extended Data Figure 4b are too small and impossible to read, and therefore the curves cannot be fully evaluated. The same applies to the FSC curves shown in Extended Data Fig. 11

Page 7, line 16: "Subsequent unambiguously docking of the Tra1 model and the crystal structures of actin-Arp4-Swr1 HSA (PDBID: 5I9E) into the map resulted in high correlation coefficients". What are the values? Could a detailed figure of the fitting be shown?

Page 7, lines 18-21: "We homology modeled the HSA and SANT domains of Eaf1 and the additional EM density at the Neck region could only accommodate ~400 additional amino acids of the Eaf1, which is largely disordered and could not allow model building at current resolution". What is the basis and level of confidence for the assignment of the HSA and SANT domains of Eaf1?

Page 7, lines 23-24: "and the main-chains of Eaf5 could be clearly traced guided by the secondary structure prediction". Could this be shown in a figure panel, either in the main figures or as Extended Data?

Page 9, lines 11-21. It would be nice to see in the paper a close up of the cryo-EM map (not just the model) showing the region corresponding to the pseudokinase domain and extended activation loop.

Page 12, lines 12-13: "Subsequent unambiguously docking of the piccolo crystal structure (PDB ID:5J9U) into the cryo-EM map resulted in high correlation coefficients.". What are the values? Could a detailed figure of the fitting be shown?

Page 12, lines 14-15: "Compared with the NuA4 TEEAA assembly, the Eaf1 conformation is much more contractive after association with piccolo (Fig. 4b)". Figure 4b shows that the differences in the NuA4 structures in the presence and absence of piccolo extensively extend to all regions, not just the Eaf1 subunit. The authors must discuss this in more detail.

Figure 1d. The cryo-EM maps cannot be seen.

Extended Data Figure 1b. The tilt axis between the two images is vertical, not horizontal as shown. The two images should be shown side-by-side with the vertical tilt axis between them. The two images could also be enlarged in order to be better appreciated.

Extended Data Figure 2. Both panels can be enlarged to be better appreciated.

Extended Data Figure 5. The cryo-EM maps are almost invisible.

Extended Data Figure 12a. The labels for repeats are very difficult to read.

Materials and Methods, page 2, line 16: "The NuA4 complex was diluted to final concentration 20-50 µg/ml". Are these values correct? They sound a little too low, even for the preparation of grids layered with continuous carbon.

Materials and Methods, page 2, lines 21-25: "The grids were firstly loaded into a Gatan 626 cryo-holder and transferred to an FEI Tecnai TF20 electron microscope to check the quality of the sample vitrification. Then, transferred to Titan Krios equipped with a field 24 emission source and operated at 300 kV." Where exactly the same grids transferred from the TF20 to the Krios, or grids prepared under similar conditions?

Materials and Methods, page 3, line 16: "the electron density map". The densities recovered by cryo-EM do not correspond to electron densities. Therefore I suggest this should read either "the map" or "the densities of the cryo-EM map".

Materials and Methods, page 3, lines 22-24: "The sequence assignment of Eaf1 HSA domain was 22 substituted in COOT guided by the clearly resolved EM densities of the bulky 23 residues in this region such as Phe, Tyr, Trp and Lys.". Could this be shown in a figure panel?

The current version of the manuscript is divided in Introduction and Results. It could be interesting to have a very brief Discussion at the end of the manuscript summarising the main results and their consequences in terms of better understanding NuA4 function.

In places there are some issues with the grammar used, but I believe those can be resolved at

editorial level.

Reviewer #2 (Remarks to the Author):

This cryo-EM manuscript presents a new structural model of the NuA4 complex – with and without the piccolo module (Esa1/Epl1/Yng2/Eaf6). The model they present is surprisingly different from the Chittuluru et al Nature Structural and Molecular Biology 2011 publication. The authors indicate that this was due to the fact that the previous published Cryo-EM structure was just that of the Tra1 subunit – a key finding supported by the recent study by Diaz-Santin eLife 2017. Not only does this study present a “holistic” model of the NuA4 complex, it provides a wealth of testable hypothesis about the architecture of this highly conserved HAT.

I appreciate the challenge and importance of generating Cryo-EM structures of the smaller state I (TEEAA-Tra1, Eaf1, Eaf5, Actin and Arp1) and larger state II (TEEAA + piccolo). However given the rocky history the NuA4 complex has with Cryo-EM and the recent Cryo-EM structure on Tra1 (Diaz-Santin eLife 2017), this reviewer would like to see at one of the many testable structure predictions actually tested.

The most intriguing observation is the TEEAA subcomplex. For example can the TEEAA subcomplex be found in any detectable amounts in the cell? Or is this also an artifact of the purification. The recent TINTIN (Eaf5/7/3) studies were able to detect this submodule independent of the larger NuA4. This is an important biological question that should be addressed especially since the authors state (page 7 line 27) “The NuA4 TEEAA, lacking the piccolo, Eaf3/7 and Eaf2/Yaf9, represents the predominant compositional state of the NuA4 complex (Fig 1d). This observation is consistent with the dissociable nature of the piccolo and the Eaf3/5/7 triad”. Which made me question whether there are TINTINs and TEEAAs complexes occurring at the same time? Or are there TEEAAs and only Eaf7/3s subcomplexes in the cell? Which brings us back to the question if the TEEAA is an artifact of the purification. If it is an artifact, it does not mean necessarily the structural information is in correct, just means the structure needs to be put in better context. To this reviewer, this is a critical question if TEEAA subcomplex is to be sold as part of the NuA4 assembly. Indeed, as the paper is presently written it suggests that TEEAA is a “real” complex.

It would also be interesting to see if any of the structural/assembly predictions are indeed true. For example mutating the key hydrophobic residues of Eaf1 HSA would test if this does in fact mediate the interaction with Act1 and Arp4 (Figures 3c and 3d). Or directly testing if the tra1-F3744A and/or tra1-F3733A disrupt Tra1-Eaf1 interaction. Or Eaf5 interaction with Eaf1 and Tra1. Confirming one or more of these predictions would significantly strengthen this study.

Secondary/Minor issues:

Page 5 lines 22-24. The authors states “Surprisingly, the sole cryo-EM reconstruction of yeast NuA4 complex was recently identified to be only Tra1 structure 29,42.” The references are incorrect and confusing. To my knowledge it was actually the Diaz-Santin eLife 2017 publication that compared the 2D class average of Tra1 with that of the 2D class average of NuA4 (Chittuluru NSMB 2011) which suggested that “the entirety of NuA4 closely matches Tra1 in appearance, indicating that the remaining NuA4 subunits are highly dynamic and/or have dissociated in the reconstruction.” Authors need to clearly clarify in the text the evidence that the cryo-EM structure of NuA4 presented by Chittuluru et al is in fact incorrect. This is important as it really is the motivation for this study.

Authors should re-proof manuscript for missing small words and/or punctuation. Egs:

Line 21/22 “There are at least two independent NuA4 subcomplexes THAT exist in vivo: ...”

Page 6 line 12/13 "Two compositional states were readily identified**,** the reference-free alignment and **the** classification

Reviewer #1 (Remarks to the Author):

The structural characterisation of NuA4 is required to fully understand the mechanisms of this critical cell regulator. In their manuscript Wang et al describe cryo-EM structures of two NuA4 assemblies. These are relevant results, but there are some issues needing some attention.

Thanks for the positive appraisal.

Page 5, lines 9-12: “past structural analyses have been limited to low resolution structures that suffer from deformation artifacts induced by chemically crosslinking and/or by staining with heavy metals”. The authors may consider revising this statement. While the previous structures are at lower resolutions, is there any direct evidence they are distorted as suggested?

Since no direct evidence suggested the previous low resolution was due to deformation artifacts, we agree with the reviewer’s suggestion and revise the statement as “past structural analyses have been limited to low resolution” in Page 5, line 11-12.

Page 6, lines 6-7: “Thus, we obtained highly homogeneous NuA4 complex suitable for cryo-EM analysis”. The authors need to revise this statement. It is a fact that they obtained a sample suitable for cryo-EM, but not a highly homogeneous one. The image analysis revealed compositional heterogeneity in the sample with the presence of different sub-complexes and in fact no structure with all 13 subunits was obtained.

We appreciate the reviewer’s suggestion and have revised the statement as “Thus, we obtained high-quality NuA4 complex suitable for cryo-EM analysis” in Page 6, line 6.

Page 6, lines 12-18. The authors did a random conical tilt (RCT) analysis of negatively stained sample in order to obtain a starting model for the analysis of the cryo-EM data, and indeed they obtained classes that proved suitable for this purpose. They describe the two dominant classes obtain from RCT as two different assembly states, based on a visual comparison. While this can be the case, the authors also need to discuss the possibility that the differences between the two classes can be the result of map distortions due to the missing cone and the flattening effect observed when RCT is done using images from negative stain. While the resulting maps may suffer from such distortions, they may still sufficiently accurate to make them suitable starting models for refinement.

Thanks for pointing out the question. Indeed, there are flattening artefacts during negative stain preparations and missing cone artefacts of the RCT reconstructions. The structural differences of the two RCT volumes may be related to such distortions. Just as the reviewer pointed out, the resulting maps could be still sufficiently accurate to be used as initial models for refinements. To minimize the effects of the possible distortions, the RCT model was low-pass filtered to 60 Å and used just as the starting model for the 3D classification.

Page 6, lines 18-19: “Both maps reveal a diamond ring shaped density at the base”. For clarity, this density should be labelled in the maps shown in Extended Data Fig 1c.

*Thanks for the reviewer's suggestion and we have labeled the density in **Supplementary Fig. 1C**.*

Page 6, line 29: “resulting in a 3D density map with final overall resolution of 4.7 Å”. At this resolution secondary structure elements should be clearly resolved. Unfortunately in the current version of the manuscript there are no figures to unambiguously show that this is the case. This can be solved by adding new panels with close-up views of the map clearly showing the secondary structure. Improving the figures where the maps are shown with fitted coordinates can also help to evaluate the level of detail resolved in the maps, since in the current version of the manuscript in all of such representations the maps are rendered as almost invisible transparent surfaces that do not allow appreciating how well the coordinates really fit. Finally, the labels for each of the FSC curves shown in Extended Data Figure 4b are too small and impossible to read, and therefore the curves cannot be fully evaluated. The same applies to the FSC curves shown in Extended Data Fig. 11

*Thanks for the reviewer's suggestion. We have added the close-up views of the map clearly showing the secondary structure (in **Fig. 1b-d**) and the more clearly FSC curves (in **Supplementary Fig. 4b and 13b**).*

Page 7, line 16: “Subsequent unambiguously docking of the Tra1 model and the crystal structures of actin-Arp4-Swr1 HSA (PDBID: 5I9E) into the map resulted in high correlation coefficients”. What are the values? Could a detailed figure of the fitting be shown?

*The correlation coefficient value of fitting Tra1 (EMD 3824, PDB ID: 5OJS) and fitting actin-Arp4-Swr1 HSA (PDBID: 5I9E) into our cryo-EM map is 0.85 and 0.63, respectively. We have now shown the detailed figures of the fitting of actin-Arp4-Swi1 HSA (PDB ID: 5I9E) in **Supplementary Fig. 9** and the*

*fitting of Tra1 (EMD 3824, PDB ID: 5OJS) into the NuA4 TEEAA complex in **Supplementary Fig. 5c**. We have reported the correlation coefficient values in the supplementary figure captions.*

Page 7, lines 18-21: “We homology modeled the HSA and SANT domains of Eaf1 and the additional EM density at the Neck region could only accommodate ~400 additional amino acids of the Eaf1, which is largely disordered and could not allow model building at current resolution”. What is the basis and level of confidence for the assignment of the HSA and SANT domains of Eaf1?

*The Eaf1 HSA domain was homology modeled based on the Swr1 HSA domain in the crystal structure of actin-Arp4-Swr1 HSA (PDBID: 5I9E). The Eaf1 SANT domain was homology modeled based on the NMR structure of SANT domain of human DNAJC2 (PDBID: 2M2E). The crystal structure of actin-Arp4-Swr1 HSA and the NMR structure of SANT domain were docked into the NuA4 cryo-EM structure as rigid bodies. HSA domain’s secondary structure element is a long alpha helix which is very characteristic and easy to identify. The Eaf1 SANT domain was identified to interact with Tra1 by functional assays. Moreover, the local resolution around Eaf1 HSA and SANT domains is close to 4 angstroms and some EM densities of the bulky residues in these regions such as Phe, Tyr, Trp and Lys are clearly resolved. The sequence assignment of Eaf1 HSA and SANT domain was substituted in COOT guided by the sidechain densities. We have now shown the detailed figure of the fitting in **Supplementary Fig. 10**.*

Page 7, lines 23-24: “and the main-chains of Eaf5 could be clearly traced guided by the secondary structure prediction”. Could this be shown in a figure panel, either in the main figures or as Extended Data?

*The Eaf5 model fitted into the cryo-EM density is now shown in **Supplementary Fig. 12**.*

Page 9, lines 11-21. It would be nice to see in the paper a close up of the cryo-EM map (not just the model) showing the region corresponding to the pseudokinase domain and extended activation loop.

*We appreciate the reviewer's suggestion. The close up of the cryo-EM map along with the fitted model showing the region corresponding to the pseudokinase domain and extended activation loop is now in **Fig. 3d, right panel**.*

Page 12, lines 12-13: "Subsequent unambiguously docking of the piccolo crystal structure (PDB ID:5J9U) into the cryo-EM map resulted in high correlation coefficients.". What are the values? Could a detailed figure of the fitting be shown?

*Piccolo contains two parts flexibly connected. We split the crystal structure of piccolo (PDBID: 5J9U) into two parts at the flexible linker (Epl1 aa 320) and separately fit them as two rigid bodies into the cryo-EM map. The correlation coefficient value of fitting four helices bundle part and fitting core part (Esa1 and Eaf1) into our cryo-EM map is 0.85 and 0.72, respectively. The detailed figure is included as **Supplementary Fig. 14**.*

Page 12, lines 14-15: "Compared with the NuA4 TEEAA assembly, the Eaf1 conformation is much more contractive after association with piccolo (Fig. 4b)". Figure 4b shows that the differences in the Nu4A structures in the presence and absence of piccolo extensively extend to all regions, not just the Eaf1 subunit. The authors must discuss this in more detail.

Thanks for the good suggestion. We have now discussed the flexibility of Eaf1 in more details in page 12, lines 12-19.

Figure 1d. The cryo-EM maps cannot be seen.

*We appreciate the reviewer's suggestion. The cryo-EM maps are now shown in **Fig. 1b-d**.*

Extended Data Figure 1b. The tilt axis between the two images is vertical, not horizontal as shown. The two images should be shown side-by-side with the vertical tilt axis between them. The two images could also be enlarged in order to be better appreciated.

Thanks for pointing out the error and we have made the change. We apologize for the mistake.

Extended Data Figure 2. Both panels can be enlarged to be better appreciated.

Thanks for the reviewer's suggestion. We have made the changes according to the reviewer's recommendation.

Extended Data Figure 5. The cryo-EM maps are almost invisible.

*We appreciate the reviewer's suggestion. We have decreased the transparency of the cryo-EM maps in **Supplementary Fig. 5**.*

Extended Data Figure 12a. The labels for repeats are very difficult to read.

*Thanks for the reviewer for pointing this out. We have re-organized the panels in **Supplementary Fig. 15** to make the repeats labels easier to read.*

Materials and Methods, page 2, line 16: "The NuA4 complex was diluted to final concentration 20-50 µg/ml". Are these values correct? They sound a little too low, even for the preparation of grids layered with continuous carbon.

*The endogenous NuA4 complex was purified directly from *Saccharomyces cerevisiae*. The concentration of NuA4 complex was not high enough for standard cryo-EM sample preparation with Quantifoil holey grid. We have applied a second layer of thin carbon film on the Quantifoil R2/1 grid to reduce the concentration demand for cryo-EM sample preparation.*

Materials and Methods, page 2, lines 21-25: "The grids were firstly loaded into a Gatan 626 cryo-holder and transferred to an FEI Tecnai TF20 electron microscope to check the quality of the sample vitrification. Then, transferred to Titan Krios equipped with a field 24 emission source and operated at 300 kV." Where exactly the same grids transferred from the TF20 to the Krios, or grids prepared under similar conditions?

We prepared a batch of grids under the same condition and use one or two grids to check the quality of the sample vitrification on TF20. The grid would be recovered if the vitrification looked good. Then the recovered grid along with the same batch of grids would be loaded into Titan Krios for data collection.

Materials and Methods, page 3, line 16: "the electron density map". The densities recovered by cryo-EM do not correspond to electron densities. Therefore I suggest this should read either "the map" or "the densities of the cryo-EM map".

Thanks for the suggestion. We have revised the "the electron density map" as

“the densities of the cryo-EM map” in Materials and Methods, page 3, line 16.

Materials and Methods, page 3, lines 22-24: “The sequence assignment of Eaf1 HSA domain was 22 substituted in COOT guided by the clearly resolved EM densities of the bulky 23 residues in this region such as Phe, Tyr, Trp and Lys.”. Could this be shown in a figure panel?

*We appreciate the reviewer's suggestion. The figure showing the densities of the cryo-EM map along with the corresponding model of Eaf1 HSA domain has been prepared and is shown in **Supplementary Fig. 10**.*

The current version of the manuscript is divided in Introduction and Results. It could be interesting to have a very brief Discussion at the end of the manuscript summarising the main results and their consequences in terms of better understanding NuA4 function.

Thanks for the reviewer's suggestion. We have summarized the main results and significance of our study as Concluding Remark in page 15, line 18- page 16, line 2.

In places there are some issues with the grammar used, but I believe those can be resolved at editorial level.

Thanks for the suggestions and we have tried our best to polish the language.

Reviewer #2 (Remarks to the Author):

This cyro-EM manuscript presents a new structural model of the NuA4 complex – with and without the piccolo module (Esa1/Epl1/Yng2/Eaf6). The model they present is surprisingly different from the Chittuluru et al Nature Structural and Molecular Biology 2011 publication. The authors indicate that this was due to the fact that the previous published Cyro-EM structure was just that of the Tra1 subunit – a key finding supported by the recent study by Diaz-Santin eLife 2017. Not only does this study present a “holistic” model of the NuA4 complex, it provides a wealth of testable hypothesis about the architecture of this highly conserved HAT.

I appreciate the challenge and importance of generating Cryo-EM structures of the smaller state I (TEEAA-Tra1, Eaf1, Eaf5, Actin and Arp1) and larger state II (TEEAA + piccolo). However given the rocky history the NuA4 complex has with Cyro-EM and the recent Cyro-EM structure on Tra1 (Diaz-Santin eLife 2017), this reviewer would like to see at one of the many testable structure predictions

actually tested.

We appreciate that the reviewer thinks that our work presents for the first time a real holistic model of the NuA4 complex. We thank the reviewer for his constructive comments and agree that our model provides a wealth of testable structure predictions about the architecture of the HAT complex. Therefore, we followed his suggestion and performed multiple in vitro and in vivo experiments to test a few interaction surfaces based on our high-resolution model of NuA4.

The most intriguing observation is the TEEAA subcomplex. For example can the TEEAA subcomplex be found in any detectable amounts in the cell? Or is this also an artifact of the purification.

We feel that our observation of the TEEAA subcomplex does not mean that it exists in the cell and could be due to several factors, including breakdown during sample preparation, as well as highly dynamic/flexible subunits difficult to be reconstructed by cryo-EM. Piccolo NuA4 is in very low amount in the cell compared to the full NuA4 complex and depends on growth conditions (Boudreault et al. G&D 2003). Purification through piccolo subunits, Eaf1, shared SWR1-C subunits always give similar stoichiometric NuA4 components, unlike TINTIN subunits which gives over-stoichiometric TINTIN over NuA4 components (Rossetto EMBO 2014; Auger MCB 2008; Zhang MCB 2005; Downs Mol Cell 2004; Boudreault G&D 2003 and several unpublished purifications with different tagged subunits as well as reports from other groups). In addition, a single NuA4 population is seen/detected by gel filtration or sucrose gradient (Allard EMBO J 1999; Galarnau Mol Cell 2000; etc...)

The recent TINTIN (Eaf5/7/3) studies were able to detect this submodule independent of the larger NuA4.

The Rossetto et al EMBO J 2014 report shows that TINTIN components are significantly more abundant than the NuA4 complex, explaining TINTIN independent nature on the top of the form associated with NuA4. Gel filtration analysis demonstrated the existence of the two forms. That being said we know that TINTIN is a bit less stably associated with NuA4 than other subunits/modules (based on gel filtration and apparent slightly less abundance in some preparations). Nevertheless, since both our structural models clearly contain Eaf5, the NuA4-anchoring subunit of TINTIN, this indicates that we are not imaging a TINTIN-less complex.

This is an important biological question that should be addressed especially since the authors state (page 7 line 27) "The NuA4 TEEAA, lacking the piccolo, Eaf3/7 and Eaf2/Yaf9, represents the predominant compositional state of the NuA4 complex (Fig 1d). This observation is consistent with the dissociable

nature of the piccolo and the Eaf3/5/7 triad". Which made me question whether there are TINTINs and TEEAAs complexes occurring at the same time?

As stated above, independent TINTIN does exist in the cell (and is now confirmed in mammals as well). Piccolo does also exist independently too but only in very low amount and specific growth conditions (Boudreault G&D 2003). On the other hand, an independent TEEAA subcomplex is doubtful and more likely arises during manipulation/fixation/imaging.

Or are there TEEAAs and only Eaf7/3s subcomplexes in the cell? Which brings us back to the question if the TEEAA is an artifact of the purification. If it is an artifact, it does not mean necessarily the structural information is incorrect, just means the structure needs to be put in better context. To this reviewer, this is a critical question if TEEAA subcomplex is to be sold as part of the NuA4 assembly. Indeed, as the paper is presently written it suggests that TEEAA is a "real" complex.

As stated above, we believe that an independent TEEAA subcomplex is unlikely in significant amount in vivo and is seen as an outcome of experimental steps to build the 3D model. This does not decrease the impact of the structural insight provided by the models. We feel it is similar to crystal structures when parts are not easily seen because more labile/dynamic/lacking stable structure. We also speculate that it may represent steps/snapshots of native full NuA4 assembly.

It would also be interesting to see if any of the structural/assembly predictions are indeed true. For example mutating the key hydrophobic residues of Eaf1 HSA would test if this does in fact mediate the interaction with Act1 and Arp4 (Figures 3c and 3d). Or directly testing if the tra1-F3744A and/or tra1-F3733A disrupt Tra1-Eaf1 interaction. Or Eaf5 interaction with Eaf1 and Tra1. Confirming one or more of these predictions would significantly strengthen this study.

As suggested, we tested several predicted interactions by GST pull-downs in vitro and in vivo mutation followed by NuA4 purification and western/mass spec analyses. We analyzed the association of the Eaf5/7/3-TINTIN module and found that purified TINTIN interacts with the first half of Eaf1 in vitro. We then produced different deletions within the Eaf1 scaffold subunit in vivo. This allowed us to determine that TINTIN associates with the N-terminal region of Eaf1. It also showed that Eaf1 HSA domain is critical for the association piccolo NuA4 and SWR1-C modules but not Tra1 or TINTIN. These results along the cryo-EM based model and previous work (Rossetto et al. 2014) indicate that Eaf5 anchors TINTIN to NuA4 through interaction with Eaf1 N-terminus. Phenotypic analysis of the yeast cells carrying Eaf1 deletions supports the

detected loss of subunits/modules but also suggests that some additional sequences are required for proper NuA4 assembly and function (new Fig. 6 and Supplementary Table 1).

We then mutated predicted key hydrophobic residues within Eaf1 HSA domain. As predicted, the triple mutant completely loses interaction with the SWR1-C shared module (Arp4-Act1-Swc4-Yaf9) in vivo. It also cripples interaction with the piccolo NuA4 module but retains some association (new Fig. 6f). These results suggest that piccolo NuA4 can interact with Eaf1 independently of the SWR1-C module but is greatly stabilized by it.

Finally, we investigated Tra1 association with NuA4/Eaf1. We found that Tra1 C-terminus (FRB-PI3K-FATC) binds Eaf1 second half in vitro. On the other hand, deleting Eaf1 SANT domain or mutating predicted key charged residues has only small effect on Tra1 association with the complex (now Fig 6g and Supplementary Table 1). This is likely explained by the fact that the very large Tra1 protein (400kDa) shows multiple distinct interactions within the NuA4 complex, as depicted in our 3D model.

Secondary/Minor issues:

Page 5 lines 22-24. The authors states “Surprisingly, the sole cryo-EM reconstruction of yeast NuA4 complex was recently identified to be only Tra1 structure 29,42.” The references are incorrect and confusing. To my knowledge it was actually the Diaz-Santin eLife 2017 publication that compared the 2D class average of Tra1 with that of the 2D class average of NuA4 (Chittuluru NSMB 2011) which suggested that “the entirety of NuA4 closely matches Tra1 in appearance, indicating that the remaining NuA4 subunits are highly dynamic and/or have dissociated in the reconstruction.” Authors need to clearly clarify in the text the evidence that the cyro-EM structure of NuA4 presented by Chittuluru et al is in fact incorrect. This is important as it really is the motivation for this study.

We thank the reviewer for pointing out the reference error. We have updated the two references arguing the Chittuluru’s model being mostly Tra1 in page 5, line 23.

Since Tra1 is shared by SAGA and NuA4, this observation suggested that lobe A contains not only Tra1 but also the 9 NuA4-specific subunits. A high flexibility of the subunits associated with Tra1 in NuA4 or a partial dissociation may explain that only the Tra1 module was reconstructed in this study.

Authors should re-proof manuscript for missing small words and/or punctuation. Egs:

Line 21/22 “There are at least two independent NuA4 subcomplexes **THAT** exist in vivo: ...”

Page 6 line 12/13 “Two compositional states were readily identified, the reference-free alignment and **the** classification

We are sorry for these mistakes and did our best to improve the text.

Reviewers' comments:

Reviewer #1 (Remarks to the Author):

The authors responded to all the points previously raised. However, their replies raised some new issues needing attention. Here I'm focusing only on the points I originally raised:

1) While the authors replied to all the point-by-point issues previously raised, in some cases they have done this only in the reply to the referees without making any corresponding changes in the manuscript. As an example, regarding the 3D maps determined using RCT in the preparation of ab-initio references for the refinement procedures, the authors described the two dominant classes obtained as different assembly states (Page 6, lines 12-18). I previously commented on this and wrote "While this can be the case, the authors also need to discuss the possibility that the differences between the two classes can be the result of map distortions due to the missing cone and the flattening effect observed when RCT is done using images from negative stain." On their reply to the referees the authors agreed with this comment, and discussed it appropriately, but the text in the manuscript remained unchanged. This needs to be rectified, as do the other cases where the points were addressed in the reply to the referees but not in the manuscript.

2) New Supplementary Figures 4 and 13. The authors revised the representation of the FSC curves and now their legends can be read. According to the authors, in both cases the green curves were determined with maps "without b-factor sharpen", while for the red curves the maps were "with b-factor sharpen". This cannot be correct. While b factors are frequently used to help in the interpretation of a map's high frequency information, they will not improve its resolution (b-factors enhance all high frequencies, both signal and noise). The authors must have done something else to improve their FSC curve estimates and they must explain what that was.

3) In continuation of point 2 above, the red curves shown in the new Supplementary Figures 4 and 13 appear artefactual and the details of the maps shown in the manuscript figures appear to be more compatible with the resolution estimates obtained from the green curves. The authors may therefore consider revising their resolution estimates, which would make the manuscript more accurate without interfering at all with any of the current interpretations of the maps or any conclusion drawn from them.

4) New Supplementary Figure 10. The authors wrote that they used the "clearly resolved" densities for the side chains of bulky residues (Phe, Tyr, Trp and Lys) to guide the sequence assignment of Eaf1 domains, which they represent in this new figure as requested. However, the details revealed in this new figure clearly indicate that there is not enough information in the map to allow such sequence assignment. The helical pitch of the helices is not resolved and therefore any assignment of densities to aminoacid side chains is speculative. A resolution at least closer to 4Å would be needed for an unbiased sequence assignment, which the maps shown clearly don't have. Therefore the following sentence should be removed from the manuscript: Materials and Methods, page 3, lines 22-14: "The sequence assignment of Eaf1 HSA domain was substituted in COOT guided by the clearly resolved EM densities of the bulky residues in this region such as Phe, Tyr, Trp and Lys."

5) New Figure 2b, new Figure 3d, new Figure 4b, new Figure 5c-d and new Supplementary Figure 12. The transparency of the maps should be reduced, in order to make them visible and also to allow the fitting of the models to be readily assessed. In the current version of the manuscript this was only done for the new Supplementary Figure 5. The same should be done for the other figures mentioned.

6) Figure 3d. The cryo-EM densities are shown on the right panel, not on the left one as currently seen

in the figure legend. Furthermore, it is not clear why the two views of the same region are represented differently in these panels. The cryo-EM densities should be shown for both views.

7) Materials and Methods, page 2, lines 21-25. As for point 1), the authors only clarified their transfer of grids between the FEI Tecnai TF20 and Krios microscopes on their reply to the referees. This must be incorporated into the manuscript, as the current text is misleading.

Reviewer #2 (Remarks to the Author):

This revised manuscript answers all my previous concerns. Further the addition of the validations of the proposed model (largely Figure 6) dramatically strengthens the story. In particular, this study begins to unravel the mystery of the Eaf1 subunit's multiple phenotypes. Highly interesting.

This study will provide the community with a "platform" to begin to understand the complex structure of NuA4/Tip60 KAT and more importantly the myriad of post-translational modifications and mutations associated with Tip60 in various diseases.

Reviewers' comments:

Reviewer #1 (Remarks to the Author):

The authors responded to all the points previously raised. However, their replies raised some new issues needing attention. Here I'm focusing only on the points I originally raised:

1) While the authors replied to all the point-by-point issues previously raised, in some cases they have done this only in the reply to the referees without making any corresponding changes in the manuscript. As an example, regarding the 3D maps determined using RCT in the preparation of ab-initio references for the refinement procedures, the authors described the two dominant classes obtained as different assembly states (Page 6, lines 12-18). I previously commented on this and wrote "While this can be the case, the authors also need to discuss the possibility that the differences between the two classes can be the result of map distortions due to the missing cone and the flattening effect observed when RCT is done using images from negative stain." On their reply to the referees the authors agreed with this comment, and discussed it appropriately, but the text in the manuscript remained unchanged. This needs to be rectified, as do the other cases where the points were addressed in the reply to the referees but not in the manuscript.

*We appreciate the reviewer's suggestion and have made the corresponding changes in the manuscript, including adding the statement of using RCT as initial models in **Page 6, lines 20-24**. In addition, we have checked all the reviewers' previous comments and have incorporated the detail regarding the procedure of the transfer of EM grids in **Materials and Methods, page 2, line 22-26**.*

2) New Supplementary Figures 4 and 13. The authors revised the representation of the FSC curves and now their legends can be read. According to the authors, in both cases the green curves were determined with maps "without b-factor sharpen", while for the red curves the maps were "with b-factor sharpen". This cannot be correct. While b factors are frequently used to help in the interpretation of a map's high frequency information, they will not improve its resolution (b-factors enhance all high frequencies, both signal and noise). The authors must have done something else to improve their FSC curve estimates and they must explain what that was.

We accepted this suggestion and changed the legend accordingly. RELION was used for data processing. After auto-refine procedures, "postprocessing" procedure was used to sharpen the map. Because the gold-standard FSC curves inside the auto-refine only use unmasked maps to prevent overfitting.

*This also means that the actual resolution is under-estimated. Postprocessing procedure is a combination of performing mask, sharpening and filtering the map. B-factor can sharpen the map but does not affect the resolution of the reconstruction. In our study, the resolution before and after postprocessing is 7.3 Å and 4.7 Å for the TEEAA assembly and 8.9 Å and 7.6 Å for the NuA4 TEEAA-piccolo assembly, respectively. The boost of resolution was caused by performing the soft mask, not B-factor. The labels “without b-factor sharpen” and “with b-factor sharpen” were changed to “Before Postprocessing” and “After Postprocessing”, respectively (in **Supplementary Fig. 4b and 13b**).*

3) In continuation of point 2 above, the red curves shown in the new Supplementary Figures 4 and 13 appear artefactual and the details of the maps shown in the manuscript figures appear to be more compatible with the resolution estimates obtained from the green curves. The authors may therefore consider revising their resolution estimates, which would make the manuscript more accurate without interfering at all with any of the current interpretations of the maps or any conclusion drawn from them.

As stated above, “postprocessing” procedure was used to sharpen the map after auto-refine procedure. The gold-standard FSC curves inside the auto-refine only use unmasked maps to prevent overfitting. This also means that the actual resolution is under-estimated. Combination of performing mask, sharpening and filtering the map which refers to “Postprocessing” procedure will lead to the boost of resolution. The local resolution analysis indicates that most of the map of TEEAA assembly is in the 4-5.2 Angstrom resolution range, with only the tip of the Head region and the outer peripheral part in the helical solenoid showing lower resolution features, which agrees well the estimated global resolution of 4.7 Angstroms.

4) New Supplementary Figure 10. The authors wrote that they used the “clearly resolved” densities for the side chains of bulky residues (Phe, Tyr, Trp and Lys) to guide the sequence assignment of Eaf1 domains, which they represent in this new figure as requested. However, the details revealed in this new figure clearly indicate that there is not enough information in the map to allow such sequence assignment. The helical pitch of the helices is not resolved and therefore any assignment of densities to aminoacid side chains is speculative. A resolution at least closer to 4Å would be needed for an unbiased sequence assignment, which the maps shown clearly don't have. Therefore the following sentence should be removed from the manuscript: Materials and Methods, page 3, lines 22-14: “The sequence assignment of Eaf1 HSA domain was substituted in COOT guided by the clearly resolved EM densities of the bulky residues in this region such as Phe, Tyr, Trp and Lys.”

*As suggested, we have revised the statement in **Materials and Methods**,*

pages 3, lines 24-25.

5) New Figure 2b, new Figure 3d, new Figure 4b, new Figure 5c-d and new Supplementary Figure 12. The transparency of the maps should be reduced, in order to make them visible and also to allow the fitting of the models to be readily assessed. In the current version of the manuscript this was only done for the new Supplementary Figure 5. The same should be done for the other figures mentioned.

*As suggested, the transparency of the cryo-EM maps have been reduced in the figures of **Figure 2b, Figure 3d, Figure 4b, Figure 5c-d and Supplementary Figure 12b.***

6) Figure 3d. The cryo-EM densities are shown on the right panel, not on the left one as currently seen in the figure legend. Furthermore, it is not clear why the two views of the same region are represented differently in these panels. The cryo-EM densities should be shown for both views.

*As suggested, the cryo-EM densities are shown for both panels in **Figure 3d.***

7) Materials and Methods, page 2, lines 21-25. As for point 1), the authors only clarified their transfer of grids between the FEI Tecnai TF20 and Krios microscopes on their reply to the referees. This must be incorporated into the manuscript, as the current text is misleading.

*Thanks for the reviewer's suggestion. We have revised the procedures of the transfer of EM grids in **Materials and Methods, page 2, line 22-26.***

Reviewer #2 (Remarks to the Author):

This revised manuscript answers all my previous concerns. Further the addition of the validations of the proposed model (largely Figure 6) dramatically strengthens the story. In particular, this study begins to unravel the mystery of the Eaf1 subunit's multiple phenotypes. Highly interesting.

This study will provide the community with a "platform" to begin to understand the complex structure of NuA4/Tip60 KAT and more importantly the myriad of post-translational modifications and mutations associated with Tip60 in various diseases.

Thanks for the positive appraisal.